# Contrastive Divergence Learning is a Time Reversal Adversarial Game

**Yair Omer**
Department of Electrical Engineering
Technion - Israel Institute of Technology
Haifa, Israel
omeryair@gmail.com

**Tomer Michaeli**
Department of Electrical Engineering
Technion - Israel Institute of Technology
Haifa, Israel
tomer.m@ee.technion.ac.il

## Abstract

Contrastive divergence (CD) learning is a classical method for fitting unnormalized statistical models to data samples. Despite its wide-spread use, the convergence properties of this algorithm are still not well understood. The main source of difficulty is an unjustified approximation which has been used to derive the gradient of the loss. In this paper, we present an alternative derivation of CD that does not require any approximation and sheds new light on the objective that is actually being optimized by the algorithm. Specifically, we show that CD is an adversarial learning procedure, where a discriminator attempts to classify whether a Markov chain generated from the model has been time-reversed. Thus, although predating generative adversarial networks (GANs) by more than a decade, CD is, in fact, closely related to these techniques. Our derivation settles well with previous observations, which have concluded that CD's update steps cannot be expressed as the gradients of any fixed objective function. In addition, as a byproduct, our derivation reveals a simple correction that can be used as an alternative to Metropolis-Hastings rejection, which is required when the underlying Markov chain is inexact (*e.g.,* when using Langevin dynamics with a large step).

## 1 Introduction

Unnormalized probability models have drawn significant attention over the years. These models arise, for example, in energy based models, where the normalization constant is intractable to compute, and are thus relevant to numerous settings. Particularly, they have been extensively used in the context of restricted Boltzmann machines (Smolensky, 1986; Hinton, 2002), deep belief networks (Hinton et al., 2006; Salakhutdinov & Hinton, 2009), Markov random fields (Carreira-Perpinan & Hinton, 2005; Hinton & Salakhutdinov, 2006), and recently also with deep neural networks (Xie et al., 2016; Song & Ermon, 2019; Du & Mordatch, 2019; Grathwohl et al., 2019; Nijkamp et al., 2019).

Fitting an unnormalized density model to a dataset is challenging due to the missing normalization constant of the distribution. A naive approach is to employ approximate maximum likelihood estimation (MLE). This approach relies on the fact that the likelihood's gradient can be approximated using samples from the model, generated using Markov Chain Monte Carlo (MCMC) techniques. However, a good approximation requires using very long chains and is thus impractical. This difficulty motivated the development of a plethora of more practical approaches, like score matching (Hyvärinen, 2005), noise contrastive estimation (NCE) (Gutmann & Hyvärinen, 2010), and conditional NCE (CNCE) (Ceylan & Gutmann, 2018), which replace the log-likelihood loss with objectives that do not require the computation of the normalization constant or its gradient.

Perhaps the most popular method for learning unnormalized models is contrastive divergence (CD) (Hinton, 2002). CD's advantage over MLE stems from its use of short Markov chains initialized at the data samples. CD has been successfully used in a wide range of domains, including modeling images (Hinton et al., 2006), speech (Mohamed & Hinton, 2010), documents (Hinton & Salakhutdinov, 2009), and movie ratings (Salakhutdinov et al., 2007), and is continuing to attract significant research attention (Liu & Wang, 2017; Gao et al., 2018; Qiu et al., 2019).

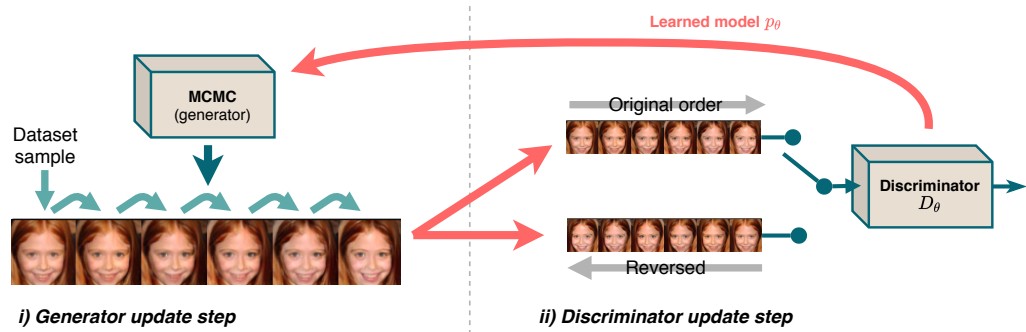

Figure 1: Contrastive divergence as an adversarial process. In the first step, the distribution model is used to generate an MCMC process which is used to generate a chain of samples. In the second step the distribution model is updated using a gradient descent step, using the MCMC transition rule.

Despite CD's popularity and empirical success, there still remain open questions regarding its theoretical properties. The primary source of difficulty is an unjustified approximation used to derive its objective's gradient, which biases its update steps (Carreira-Perpinan & Hinton, 2005; Bengio & Delalleau, 2009). The difficulty is exacerbated by the fact that CD's update steps cannot be expressed as the gradients of any fixed objective (Tieleman, 2007; Sutskever & Tieleman, 2010).

In this paper, we present an alternative derivation of CD, which relies on completely different principles and requires no approximations. Specifically, we show that CD's update steps are the gradients of an adversarial game in which a discriminator attempts to classify whether a Markov chain generated from the model is presented to it in its original or a time-reversed order (see Fig. 1). Thus, our derivation sheds new light on CD's success: Similarly to modern generative adversarial methods (Goodfellow et al., 2014), CD's discrimination task becomes more challenging as the model approaches the true distribution. This keeps the update steps effective throughout the entire training process and prevents early saturation as often happens in non-adaptive methods like NCE and CNCE. In fact, we derive CD as a natural extension of the CNCE method, replacing the fixed distribution of the contrastive examples with an adversarial adaptive distribution.

CD requires that the underlying MCMC be exact, which is not the case for popular methods like Langevin dynamics. This commonly requires using Metropolis-Hastings (MH) rejection, which ignores some of the generated samples. Interestingly, our derivation reveals an alternative correction method for inexact chains, which does not require rejection.

## 2 BACKGROUND

### 2.1 THE CLASSICAL DERIVATION OF CD

Assume we have an unnormalized distribution model $p_\theta$. Given a dataset of samples $\{x_i\}$ independently drawn from some unknown distribution $p$, CD attempts to determine the parameters $\theta$ with which $p_\theta$ best explains the dataset. Rather than using the log-likelihood loss, CD's objective involves distributions of samples along finite Markov chains initialized at $\{x_i\}$. When based on chains of length $k$, the algorithm is usually referred to as CD-$k$.

Concretely, let $q_\theta(x'|x)$ denote the transition rule of a Markov chain with stationary distribution $p_\theta$, and let $r_\theta^m$ denote the distribution of samples after $m$ steps of the chain. As the Markov chain is initialized from the dataset distribution and converges to $p_\theta$, we have that $r_\theta^0 = p$ and $r_\theta^\infty = p_\theta$. The CD algorithm then attempts to minimize the loss

$$\ell_{\text{CD-}k} = D_{\text{KL}}(r_\theta^0 || r_\theta^\infty) - D_{\text{KL}}(r_\theta^k || r_\theta^\infty)$$
$$= D_{\text{KL}}(p || p_\theta) - D_{\text{KL}}(r_\theta^k || p_\theta), \tag{1}$$

where $D_{\text{KL}}$ is the Kullback-Leibler divergence. Under mild conditions on $q_\theta$ (Cover & Halliwell, 1994) this loss is guaranteed to be positive, and it vanishes when $p_\theta = p$ (in which case $r_\theta^k = p_\theta$).

To allow the minimization of (1) using gradient-based methods, one can write

$$\nabla_\theta \ell_{\text{CD-}k} = \mathbb{E}_{\tilde{X} \sim r_\theta^k}[\nabla_\theta \log p_\theta(\tilde{X})] - \mathbb{E}_{X \sim p}[\nabla_\theta \log p_\theta(X)] + \frac{dD_{\text{KL}}(r_\theta^k || p_\theta)}{dr_\theta^k} \nabla_\theta r_\theta^k. \qquad (2)$$

Here, the first two terms can be approximated using two batches of samples, one drawn from $p$ and one from $r_\theta^k$. The third term is the derivative of the loss with respect only to the $\theta$ that appears in $r_\theta^k$, ignoring the dependence of $p_\theta$ on $\theta$. This is the original notation from (Hinton, 2002); an alternative way to write this term would be $\nabla_{\tilde{\theta}} D_{\text{KL}}(r_{\tilde{\theta}}^k || p_\theta)$. This term turns out to be intractable and in the original derivation, it is argued to be small and thus neglected, leading to the approximation

$$\nabla_\theta \ell_{\text{CD-}k} \approx \frac{1}{n} \sum_i \left( \nabla_\theta \log p_\theta(\tilde{x}_i) - \nabla_\theta \log p_\theta(x_i) \right) \qquad (3)$$

Here $\{x_i\}$ is a batch of $n$ samples from the dataset and $\{\tilde{x}_i\}$ are $n$ samples generated by applying $k$ MCMC steps to each of the samples in that batch. The intuition behind the resulting algorithm (summarized in App. A) is therefore simple. In each gradient step $\theta \leftarrow \theta - \eta \nabla_\theta \ell_{\text{CD-}k}$, the log-likelihood of samples from the dataset is increased on the expense of the log-likelihood of the contrastive samples $\{\tilde{x}_i\}$, which are closer to the current learned distribution $p_\theta$.

Despite the simple intuition, it has been shown that without the third term, CD's update rule (2) generally cannot be the gradient of any fixed objective (Tieleman, 2007; Sutskever & Tieleman, 2010) except for some very specific cases. For example, Hyvärinen (2007) has shown that when the Markov chain is based on Langevin dynamics with a step size that approaches zero, the update rule of CD-1 coincides with that of score-matching Hyvärinen (2005). Similarly, the probability flow method of Sohl-Dickstein et al. (2011) has been shown to be equivalent to CD with a very unique Markov chain. Here, we show that regardless of the selection of the Markov chain, the update rule is in fact the exact gradient of a particular adversarial objective, which adapts to the current learned model in each step.

## 2.2 Conditional Noise Contrastive Estimation

Our derivation views CD as an extension of the CNCE method, which itself is an extension of NCE. We therefore start by briefly reviewing those two methods.

In NCE, the unsupervised density learning problem is transformed into a supervised one. This is done by training a discriminator $D_\theta(x)$ to distinguish between samples drawn from $p$ and samples drawn from some preselected contrastive distribution $p_{\text{ref}}$. Specifically, let the random variable $Y$ denote the label of the class from which the variable $X$ has been drawn, so that $X|(Y = 1) \sim p$ and $X|(Y = 0) \sim p_{\text{ref}}$. Then it is well known that the discriminator minimizing the binary cross-entropy (BCE) loss is given by

$$D_{\text{opt}}(x) = \mathbb{P}(Y = 1 | X = x) = \frac{p(x)}{p(x) + p_{\text{ref}}(x)}. \qquad (4)$$

Therefore, letting our parametric discriminator have the form

$$D_\theta(x) = \frac{p_\theta(x)}{p_\theta(x) + p_{\text{ref}}(x)}, \qquad (5)$$

and training it with the BCE loss, should in theory lead to $D_\theta(x) = D_{\text{opt}}(x)$ and thus to $p_\theta(x) = p(x)$. In practice, however, the convergence of NCE highly depends on the selection of $p_{\text{ref}}$. If it significantly deviates from $p$, then the two distributions can be easily discriminated even when the learned distribution $p_\theta$ is still very far from $p$. At this point, the optimization essentially stops updating the model, which can result in a very inaccurate estimate for $p$. In the next section we provide a precise mathematical explanation for this behavior.

The CNCE method attempts to alleviate this problem by drawing the contrastive samples based on the dataset samples. Specifically, each dataset sample $x$ is paired with a contrastive sample $\tilde{x}$ that is drawn conditioned on $x$ from some predetermined conditional distribution $q(\tilde{x}|x)$ (e.g. $\mathcal{N}(x, \sigma^2 I)$). The pair is then concatenated in a random order, and a discriminator is trained to predict the correct

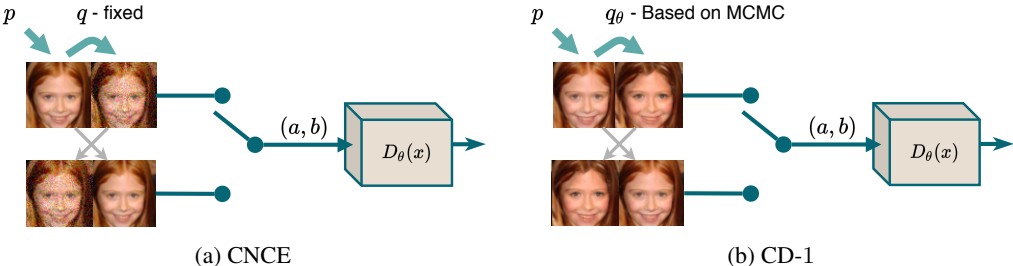

(a) CNCE                                      (b) CD-1

Figure 2: **From CNCE to CD-**1. (a) In CNCE, each contrastive sample is generated using a fixed conditional distribution $q(\cdot|\cdot)$ (which usually corresponds to additive noise). The real and fake samples are then concatenated and presented to a discriminator in a random order, which is trained to predict the correct order. (b) CD-1 can be viewed as CNCE with a $q(\cdot|\cdot)$ that corresponds to the transition rule of a Markov chain with stationary distribution $p_\theta$. Since $q$ depends on $p_\theta$ (hence the subscript $\theta$), during training the distribution of contrastive samples becomes more similar to that of the real samples, making the discrimination task harder.

order. This is illustrated in Fig. 2a. Specifically, here the two classes are of pairs $(A, B)$ corresponding to $(A, B) = (X, \tilde{X})$ for $Y = 1$, and $(A, B) = (\tilde{X}, X)$ for $Y = 0$, and the discriminator minimizing the BCE loss is given by

$$D_{\text{opt}}(a, b) = \mathbb{P}(Y = 1 | A = a, B = b) = \frac{q(b|a)p(a)}{q(b|a)p(a) + q(a|b)p(b)}. \tag{6}$$

Therefore, constructing a parametric discriminator of the form

$$D_\theta(a, b) = \frac{q(b|a)p_\theta(a)}{q(b|a)p_\theta(a) + q(a|b)p_\theta(b)} = \left(1 + \frac{q(a|b)p_\theta(b)}{q(b|a)p_\theta(a)}\right)^{-1}, \tag{7}$$

and training it with the BCE loss, should lead to $p_\theta \propto p$. Note that here $D_\theta$ is indifferent to a scaling of $p_\theta$, which is thus determined only up to an arbitrary multiplicative constant.

CNCE improves upon NCE, as it allows working with contrastive samples whose distribution is closer to $p$. However, it does not completely eliminate the problem, especially when $p$ exhibits different scales of variation in different directions. This is the case, for example, with natural images, which are known to lie close to a low-dimensional manifold. Indeed if the conditional distribution $q(\cdot|\cdot)$ is chosen to have a small variance, then CNCE fails to capture the global structure of $p$. And if $q(\cdot|\cdot)$ is taken to have a large variance, then CNCE fails to capture the intricate features of $p$ (see Fig. 3). The latter case can be easily understood in the context of images (see Fig. 2a). Here, the discriminator can easily distinguish which of its pair of input images is the noisy one, without having learned an accurate model for the distribution of natural images (*e.g.,* simply by comparing their smoothness). When this point is reached, the optimization essentially stops.

In the next section we show that CD is in fact an adaptive version of CNCE, in which the contrastive distribution is constantly updated in order to keep the discrimination task hard. This explains why CD is less prone to early saturation than NCE and CNCE.

## 3 AN ALTERNATIVE DERIVATION OF CD

We now present our alternative derivation of CD. In Sec. 3.1 we identify a decomposition of the CNCE loss, which reveals the term that is responsible for early saturation. In Sec. 3.2, we then present a method for adapting the contrastive distribution in a way that provably keeps this term bounded away from zero. Surprisingly, the resulting update step turns out to precisely match that of CD-1, thus providing a new perspective on CD learning. In Sec. 3.3, we extend our derivation to include CD-$k$ (with $k \geq 1$).

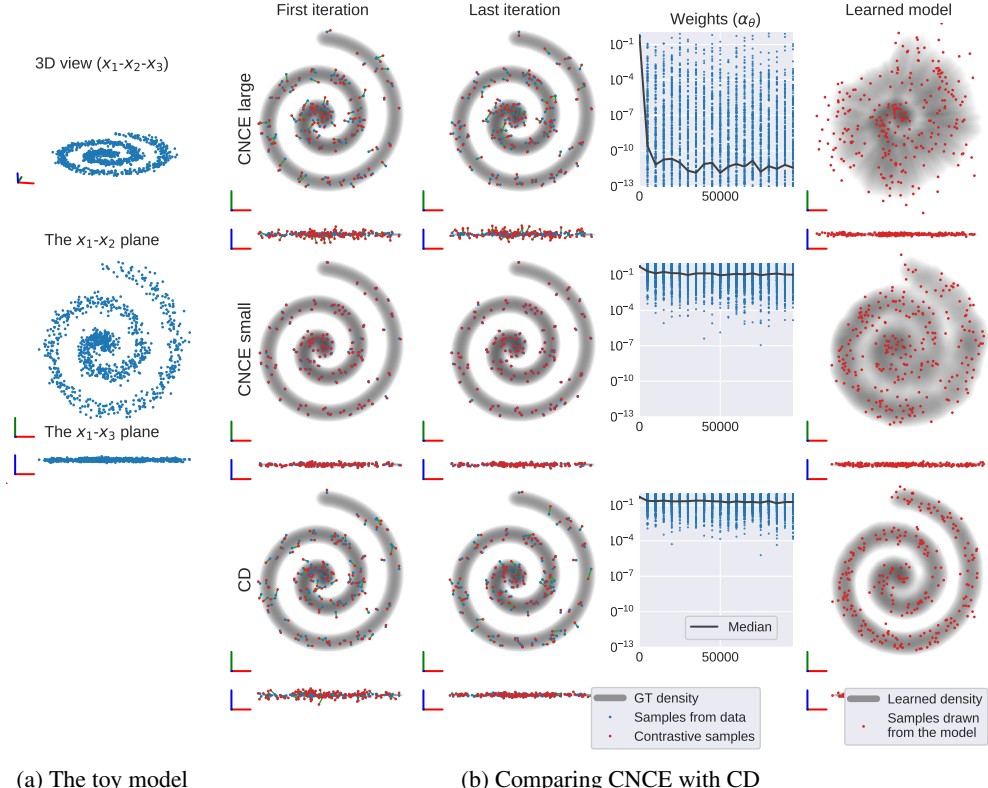

(a) The toy model  (b) Comparing CNCE with CD

Figure 3: A toy example illustrating the importance of the adversarial nature of CD. Here, the data lies close to a 2D spiral embedded in a 10-dimensional space. (a) The training samples in the first 3 dimensions. (b) Three different approaches for learning the distribution: CNCE with large contrastive variance (top), CNCE with small contrastive variance (middle), and CD based on Langevin dynamics MCMC with the weight adjustment described in Sec. 3.4 (bottom). As can be seen in the first two columns, CD adapts the contrastive samples according to the data distribution, whereas CNCE does not. Therefore, CNCE with large variance fails to learn the distribution because the vast majority of its contrastive samples are far from the manifold and quickly become irrelevant (as indicated by the weights $\alpha_\theta$ in the third column). And CNCE with small variance fails to learn the global structure of the distribution because its contrastive samples are extremely close to the dataset samples. CD, on the other hand, adjusts the contrastive distribution during training, so as to generate samples that are close to the manifold yet traverse large distances along it.

## 3.1 REINTERPRETING CNCE

Let us denote

$$w_\theta(a, b) \triangleq \frac{q(a|b)p_\theta(b)}{q(b|a)p_\theta(a)}, \tag{8}$$

so that we can write CNCE's discriminator (7) as

$$D_\theta(a, b) = (1 + w_\theta(a, b))^{-1}. \tag{9}$$

Then we have the following observation (see proof in App. B).

**Observation 1.** *The gradient of the CNCE loss can be expressed as*

$$\nabla_\theta \ell_{CNCE} = \mathbb{E}_{\substack{X \sim p \\ \tilde{X}|X \sim q}} \left[ \alpha_\theta(X, \tilde{X}) \left( \nabla_\theta \log p_\theta(\tilde{X}) - \nabla_\theta \log p_\theta(X) \right) \right], \tag{10}$$

*where*

$$\alpha_\theta(x, \tilde{x}) \triangleq (1 + w_\theta(x, \tilde{x})^{-1})^{-1}. \tag{11}$$

Note that (10) is similar in nature to the (approximate) gradient of the CD loss (3). Particularly, as in CD, the term $\nabla_\theta \log p_\theta(\tilde{X}) - \nabla_\theta \log p_\theta(X)$ causes each gradient step to increase the log-likelihood of samples from the dataset on the expense of the log-likelihood of the contrastive samples. However, as opposed to CD, here we also have the coefficient $\alpha_\theta(x, \tilde{x})$, which assigns a weight between 0 and 1 to each pair of samples $(x, \tilde{x})$. To understand its effect, observe that

$$\alpha_\theta(x, \tilde{x}) = 1 - D_\theta(x, \tilde{x}) = D_\theta(\tilde{x}, x). \tag{12}$$

Namely, this coefficient is precisely the probability that the discriminator assigns to the incorrect order of the pair. Therefore, this term gives a low weight to "easy" pairs (*i.e.,* for which $D_\theta(x, \tilde{x})$ is close to 1) and a high weight to "hard" ones.

This weighting coefficient is of course essential for ensuring convergence to $p$. For example, it prevents $\log p_\theta$ from diverging to $\pm\infty$ when the discriminator is presented with the same samples over and over again. The problem is that a discriminator can often correctly discriminate all training pairs, even with a $p_\theta$ that is still far from $p$. In such cases, $\alpha_\theta$ becomes practically zero for all pairs and the model stops updating. This shows that a good contrastive distribution is one which keeps the discrimination task hard throughout the training. As we show next, there is a particular choice which provably prevents $\alpha_\theta$ from converging to zero, and that choice results in the CD method.

## 3.2 FROM CNCE TO CD-1

To bound $\alpha_\theta$ away from 0, and thus avoid the early stopping of the training process, we now extend the original CNCE algorithm by allowing the conditional distribution $q$ to depend on $p_\theta$ (and thus to change from one step to the next). Our next key observation is that in this setting there exists a particular choice that keeps $\alpha_\theta$ constant.

**Observation 2.** *If $q$ is chosen to be the transition probability of a reversible Markov chain with stationary distribution $p_\theta$, then*

$$\alpha_\theta(x, \tilde{x}) = \frac{1}{2}, \quad \forall x, \tilde{x}. \tag{13}$$

*Proof.* A reversible chain with transition $q$ and stationary distribution $p_\theta$, satisfies the detailed balance property

$$q(\tilde{x}|x)p_\theta(x) = q(x|\tilde{x})p_\theta(\tilde{x}), \quad \forall x, \tilde{x}. \tag{14}$$

Substituting (14) into (8) leads to $w_\theta(x, \tilde{x}) = 1$, which from (11) implies $\alpha_\theta(x, \tilde{x}) = \frac{1}{2}$. $\qquad \square$

This observation directly links CNCE to CD. First, the suggested method for generating the contrastive samples is precisely the one used in CD-1. Second, as this choice of $q$ leads to $\alpha_\theta(x, \tilde{x}) = \frac{1}{2}$, it causes the gradient of the CNCE loss (10) to become

$$\nabla_\theta \ell_{\text{CNCE}} = \frac{1}{2} \mathbb{E}_{\substack{X \sim p \\ \tilde{X}|X \sim q}} \left[ \nabla_\theta \log p_\theta(\tilde{X}) - \nabla_\theta \log p_\theta(X) \right], \tag{15}$$

which is exactly proportional to the CD-1 update (3). We have thus obtained an alternative derivation of CD-1. Namely, rather than viewing CD-1 learning as an *approximate* gradient descent process for the loss (1), we can view each step as the *exact* gradient of the CNCE discrimination loss, where the reference distribution $q$ is adapted to the current learned model $p_\theta$. This is illustrated in Fig. 2b.

Since $q$ is chosen based on $p_\theta$, the overall process is in fact an adversarial game. Namely, the optimization alternates between updating $q$, which acts as a generator, and updating $p_\theta$, which defines the discriminator. As $p_\theta$ approaches $p$, the distribution of samples generated from the MCMC also becomes closer to $p$, which makes the discriminator's task harder and thus prevents early saturation.

It should be noted that formally, since $q$ depends on $p_\theta$, it also indirectly depends on $\theta$, so that a more appropriate notation would be $q_\theta$. However, during the update of $p_\theta$ we fix $q_\theta$ (and vise versa), so that the gradient in the discriminator update does not consider the dependence of $q_\theta$ on $\theta$. This is why (15) does not involve the gradient of $\tilde{X}$ which depends on $q_\theta$.

The reason for fixing $q_\theta$ comes from the adversarial nature of the learning process. Being part of the chain generation process, the goal of the transition rule $q_\theta$ is to generate chains that appear to be time-reversible, while the goal of the classifier, which is based on the model $p_\theta$, is to correctly classify

whether the chains were reversed. Therefore, we do not want the optimization of the classifier to affect $q_\theta$. This is just like in GANs, where the generator and discriminator have different objectives, and so when updating the discriminator the generator is kept fixed.

### 3.3 FROM CD-1 TO CD-$k$

To extend our derivation to CD-$k$ with an arbitrary $k \geq 1$, let us now view the discrimination problem of the previous section as a special case of a more general setting. Specifically, the pairs of samples presented to the discriminator in Sec. 3.2, can be viewed as Markov chains of length two (comprising the initial sample from the dataset and one extra generated sample). It is therefore natural to consider also Markov chains of arbitrary lengths. That is, assume we initialize the MCMC at a sample $x_i$ from the dataset and run it for $k$ steps to obtain a sequence $(x^{(0)}, x^{(1)}, \ldots, x^{(k)})$, where $x^{(0)} = x_i$. We can then present this sequence to a discriminator either in its original order, or time-reversed, and train the discriminator to classify the correct order. We coin this a *time-reversal classification task*. Interestingly, in this setting, we have the following.

**Observation 3.** *When using a reversible Markov chain of length $k + 1$ with stationary distribution $p_\theta$, the gradient of the BCE loss of the time-reversal classification task is given by*

$$\nabla_\theta \ell_{CNCE} = \tfrac{1}{2} \mathbb{E} \left[ \nabla_\theta \log p_\theta(X^{(k)}) - \nabla_\theta \log p_\theta(X^{(0)}) \right], \tag{16}$$

*which is exactly identical to the CD-$k$ update (3) up to a multiplicative factor of $\tfrac{1}{2}$.*

This constitutes an alternative interpretation of CD-$k$. That is, CD-$k$ can be viewed as a time-reversal adversarial game, where in each step, the model $p_\theta$ is updated so as to allow the discriminator to better distinguish MCMC chains from their time-reversed counterparts.

Two remarks are in order. First, it is interesting to note that although the discriminator's task is to classify the order of the whole chain, its optimal strategy is to examine only the endpoints of the chain, $x^{(0)}$ and $x^{(k)}$. Second, it is insightful to recall that the original motivation behind the CD-$k$ loss (1) was that when $p_\theta$ equals $p$, the marginal probability of each individual step in the chain is also $p$. Our derivation, however, requires more than that. To make the chain indistinguishable from its time-reversed version, the joint probability of all samples in the chain must be invariant to a flip of the order. When $p_\theta = p$, this is indeed the case, due to the detailed balance property (14).

*Proof of Observation 3.* We provide the outline of the proof (see full derivation in App. C). Let $(A^{(0)}, A^{(1)}, \ldots, A^{(k)})$ denote the input to the discriminator and let $Y$ indicate the order of the chain, with $Y = 1$ corresponding to $(A^{(0)}, A^{(1)}, \ldots, A^{(k)}) = (X^{(0)}, X^{(1)}, \ldots, X^{(k)})$ and $Y = 0$ to $(A^{(0)}, A^{(1)}, \ldots, A^{(k)}) = (X^{(k)}, X^{(k-1)}, \ldots, X^{(0)})$. The discriminator that minimizes the BCE loss is now given by

$$D(a_0, a_1, \ldots, a_k) = \mathbb{P}(Y = 1 | A^{(0)} = a_0, A^{(1)} = a_1, \ldots, A^{(k)} = a_k)$$

$$= \left( 1 + \frac{q(a_0|a_1) \cdots q(a_{k-1}|a_k) p(a_k)}{q(a_k|a_{k-1}) \cdots q(a_1|a_0) p(a_0)} \right)^{-1}$$

$$= \left( 1 + \prod_{i=1}^{k} w_\theta(a_{i-1}, a_i) \right)^{-1}. \tag{17}$$

The CNCE paradigm thus defines a discriminator $D_\theta$ having the form of (17) but with $p$ replaced by $p_\theta$. Recall that despite the dependence of the transition probability $q$ on the current learned model $p_\theta$, it is regarded as fixed within each discriminator update step. We therefore omit the subscript $\theta$ from $q$ here. Similarly to the derivation of (10), explicitly writing the gradient of the BCE loss of our discriminatrion task, gives

$$\nabla_\theta \ell_{\text{chain}} = \mathbb{E} \left[ \left( 1 + \prod_{i=1}^{k} w_\theta(X^{(i-1)}, X^{(i)})^{-1} \right)^{-1} \left( \nabla_\theta \log p_\theta(X^{(k)}) - \nabla_\theta \log p_\theta(X^{(0)}) \right) \right]$$

$$= \mathbb{E} \left[ \alpha_\theta(X^{(0)}, \ldots, X^{(k)}) \left( \nabla_\theta \log p_\theta(X^{(k)}) - \nabla_\theta \log p_\theta(X^{(0)}) \right) \right]. \tag{18}$$

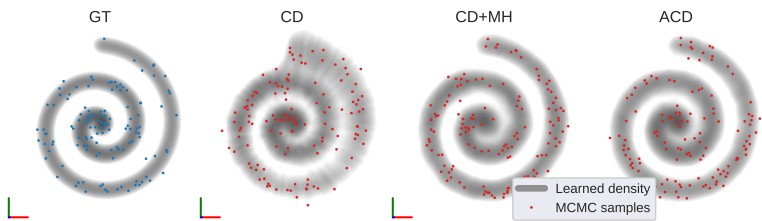

Figure 4: Here, we use different CD configurations for learning the model of Fig. 3. All configurations use Langevin dynamics as their MCMC process, but with different ways of compensating for the lack of detailed balance. From left to right we have the ground-truth density, CD w/o any correction, CD with Metropolis-Hastings rejection, and CD with our proposed adjustment.

where we now defined

$$\alpha_\theta(a_0, \ldots, a_k) \triangleq \left( 1 + \prod_{i=1}^{k} w_\theta(a_{i-1}, a_i)^{-1} \right)^{-1}. \tag{19}$$

Note that (10) is a special case of (18) corresponding to $k = 1$, where $X$ and $\tilde{X}$ in (10) are $X^{(0)}$ and $X^{(1)}$ in (18). As before, when $q$ satisfies the detailed balance property (14), we obtain $w_\theta = 1$ and consequently the weighting term $\alpha_\theta$ again equals $\frac{1}{2}$. Thus, the gradient (18) reduces to (16), which is exactly proportional to the CD-$k$ update (3). $\qquad \square$

### 3.4 MCMC Processes That do not Have Detailed Balance

In our derivation, we assumed that the MCMC process is reversible, and thus exactly satisfies the detailed balance property (14). This assumption ensured that $w_\theta = 1$ and thus $\alpha_\theta = \frac{1}{2}$. In practice, however, commonly used MCMC methods satisfy this property only approximately. For example, the popular discrete Langevin dynamics process obeys detailed balance only in the limit where the step size approaches zero. The common approach to overcome this is through Metropolis-Hastings (MH) rejection (Hastings, 1970), which guarantees detailed balance by accepting only a portion of the proposed MCMC transitions. In this approach, the probability of accepting a transition from $x$ to $\tilde{x}$ is closely related to the weighing term $w_\theta$, and is given by

$$A(x, \tilde{x}) = \min\left(1, w_\theta(x, \tilde{x})\right). \tag{20}$$

Interestingly, our derivation reveals an alternative method for accounting for lack of detailed balance.

Concretely, we saw that the general expression for the gradient of the BCE loss (before assuming detailed balance) is given by (18). This expression differs from the original update step of CD-$k$ only in the weighting term $\alpha_\theta(x^{(0)}, \ldots, x^{(k)})$. Therefore, all that is required for maintaining correctness in the absence of detailed balance, is to weigh each chain by its "hardness" $\alpha_\theta(x^{(0)}, \ldots, x^{(k)})$ (see Alg. 2 in App. A). Note that in this case, the update depends not only on the end-points of the chains, but rather also on their intermediate steps. As can be seen in Fig. 4, this method performs just as well as MH, and significantly better than vanilla CD without correction.

## 4 Illustration Through a Toy Example

To illustrate our observations, we now conclude with a simple toy example (see Fig. 3). Our goal here is not to draw general conclusions regarding the performance of CNCE and CD, but rather merely to highlight the adversarial nature of CD and its importance when the data density exhibits different scales of variation along different directions.

We take data concentrated around a 2-dimensional manifold embedded in 10-dimensional space. Specifically, let $e^{(1)}, \ldots, e^{(10)}$ denote the standard basis in $\mathbb{R}^{10}$. Then each data sample is generated by adding Gaussian noise to a random point along a 2D spiral lying in the $e^{(1)}$-$e^{(2)}$ plane. The STD of the noise in the $e^{(1)}$ and $e^{(2)}$ directions is 5 times larger than that in the other 8 axes. Figure 3a shows the projections of the the data samples onto the the first 3 dimensions. Here, we use a

multi-layer perceptron (MLP) as our parametric model, $\log p_\theta$, and train it using several different learning configurations (for the full details see App. D).

Figure 3b visualizes the training as well as the final result achieved by each configuration. The first two rows show CNCE with Gaussian contrastive distributions of two different STDs. The third row shows the adjusted CD described in Sec. 3.4 with Langevin Dynamics as its MCMC process. As can be seen, for CNCE with a large STD, the contrastive samples are able to explore large areas around the original samples, but this causes their majority to lie relatively far from the manifold (see their projections onto the $e^{(1)}$-$e^{(3)}$ plane). In this case, $\alpha_\theta$ decreases quickly, causing the learning process to ignore most samples at a very early stage of the training. When using CNCE with a small STD, the samples remain relevant throughout the training, but this comes at the price of inability to capture the global structure of the distribution. CD, on the other hand, is able to enjoy the best of both worlds as it adapts the contrastive distribution over time. Indeed, as the learning progresses, the contrastive samples move closer to the manifold to maintain their relevance. Note that since we use the adjusted version of CD, the weights in this configuration are not precisely 1. We chose the step size of the Langevin Dynamics so that the median of the weights is approximately $10^{-2}$.

Figure 4 shows the results achieved by different variants of CD. As can be seen, without correcting for the lack of detailed balance, CD fails to estimate the density correctly. When using MH rejection to correct the MCMC, or our adaptive CD (ADC) to correct the update steps, the estimate is significantly improved.

## 5 CONCLUSION

The classical CD method has seen many uses and theoretical analyses over the years. The original derivation presented the algorithm as an approximate gradient descent process for a certain loss. However, the accuracy of the approximation has been a matter of much dispute, leaving it unclear what objective the algorithm minimizes in practice. Here, we presented an alternative derivation of CD's update steps, which involves no approximations. Our analysis shows that CD is in essence an adversarial learning procedure, where a discriminator is trained to distinguish whether a Markov chain generated from the learned model has been time-flipped or not. Therefore, although predating GANs by more than a decade, CD in fact belongs to the same family of techniques. This provides a possible explanation for its empirical success.

**Acknowledgement**   This research was supported by the Technion Ollendorff Minerva Center.

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

# A ALGORITHMS

Below we summarize the algorithms of the classical CD and the proposed adjusted version described in Sec. 3.4.

---

**Algorithm 1:** Contrastive Divergence - $k$

---

**Require:** parametric model $p_\theta$, MCMC transition rule $q_\theta(\cdot|\cdot)$ with stationary distribution $p_\theta$, step size $\eta$, chain length $k$.

**while** *not converged* **do**

    Sample a batch $\{x_i\}_{i=1}^n$ from the dataset

    Initialize $\{\tilde{x}_i\}_{i=1}^n$ to be a copy of the batch

    **for** $i$=1 **to** $n$ **do**

        **for** $j$=1 **to** $k$ **do**

            Draw a sample $x'$ from $q_\theta(\cdot|\tilde{x}_i)$

            $\tilde{x}_i \leftarrow x'$

        **end**

        $g_i \leftarrow \nabla_\theta \log p_\theta(\tilde{x}_i) - \nabla_\theta \log p_\theta(x_i)$

    **end**

    $\theta \leftarrow \theta - \eta \frac{1}{n} \sum_i g_i$

**end**

---

---

**Algorithm 2:** Adjusted Contrastive Divergence - $k$

---

**Require:** parametric model $p_\theta$, MCMC transition rule $q_\theta(\cdot|\cdot)$ whose stationary distribution is $p_\theta$, step size $\eta$, chain length $k$.

**while** *not converged* **do**

    Sample a batch $\{x_i\}_{i=1}^n$ from the dataset

    Initialize $\{\tilde{x}_i\}_{i=1}^n$ to be a copy of the batch

    **for** $i$=1 **to** $n$ **do**

        $w_i^{\text{tot}} \leftarrow 1$

        **for** $j$=1 **to** $k$ **do**

            Draw a sample $x'$ from $q_\theta(\cdot|\tilde{x}_i)$

            $w_i^{\text{tot}} \leftarrow w_i^{\text{tot}} \cdot \frac{q(x_i|x')p_\theta(x')}{q(x'|x_i)p_\theta(x_i)}$

            $\tilde{x}_i \leftarrow x'$

        **end**

        $\alpha_i \leftarrow (1 + 1/w_i^{\text{tot}})^{-1}$

        $g_i \leftarrow \nabla_\theta \log p_\theta(\tilde{x}_i) - \nabla_\theta \log p_\theta(x_i)$

    **end**

    $\theta \leftarrow \theta - \eta \frac{1}{n} \sum_i \alpha_i \cdot g_i$

**end**

---

# B DERIVATION OF CNCE'S GRADIENT

*Proof of Observation 1.* The BCE loss achieved by the CNCE discriminator (7) is given by

$$\ell_{\text{CNCE}} = -\frac{1}{2}\mathbb{E}_{\substack{A\sim p \\ B|A\sim q}}\left[\log(D_\theta(A,B))\right] - \frac{1}{2}\mathbb{E}_{\substack{B\sim p \\ A|B\sim q}}\left[\log(1-D_\theta(A,B))\right] =$$

$$= -\mathbb{E}_{\substack{X\sim p \\ \tilde{X}|X\sim q}}\left[\log(D_\theta(X,\tilde{X}))\right], \tag{21}$$

where we used the fact that $1 - D_\theta(a, b) = D_\theta(b, a)$. Now, substituting the definition of $D_\theta$ form (9), the gradient of (21) can be expressed as

$$
\begin{aligned}
\nabla_\theta \ell_{\text{CNCE}} &= \mathbb{E}_{\substack{X \sim p \\ \tilde{X}|X \sim q}} \left[ \nabla_\theta \log \left( 1 + w_\theta(X, \tilde{X}) \right) \right] \\
&= \mathbb{E}_{\substack{X \sim p \\ \tilde{X}|X \sim q}} \left[ \left( 1 + w_\theta(X, \tilde{X}) \right)^{-1} \nabla_\theta w_\theta(X, \tilde{X}) \right] \\
&= \mathbb{E}_{\substack{X \sim p \\ \tilde{X}|X \sim q}} \left[ \left( 1 + w_\theta(X, \tilde{X}) \right)^{-1} \frac{w_\theta(X, \tilde{X})}{w_\theta(X, \tilde{X})} \nabla_\theta w_\theta(X, \tilde{X}) \right] \\
&= \mathbb{E}_{\substack{X \sim p \\ \tilde{X}|X \sim q}} \left[ \left( \frac{1 + w_\theta(X, \tilde{X})}{w_\theta(X, \tilde{X})} \right)^{-1} \frac{\nabla_\theta w_\theta(X, \tilde{X})}{w_\theta(X, \tilde{X})} \right] \\
&= \mathbb{E}_{\substack{X \sim p \\ \tilde{X}|X \sim q}} \left[ \left( 1 + w_\theta(X, \tilde{X})^{-1} \right)^{-1} \nabla_\theta \log(w_\theta(X, \tilde{X})) \right] \\
&= \mathbb{E}_{\substack{X \sim p \\ \tilde{X}|X \sim q}} \left[ \alpha_\theta(X, \tilde{X}) \left( \nabla_\theta \log p_\theta(\tilde{X}) - \nabla_\theta \log p_\theta(X) \right) \right], \quad (22)
\end{aligned}
$$

where we used the fact that $\nabla_\theta w_\theta = w_\theta \nabla_\theta \log(w_\theta)$ and the definition of $\alpha_\theta$ from (11). $\qquad\square$

## C    DERIVATION OF THE GRADIENT OF CNCE WITH MULTIPLE MC STEPS

We here describe the full derivation of the gradient in (18) following the same steps as in (22). The BCE loss achieved by the discriminator in (17) is given by

$$
\ell_{\text{chain}} = -\mathbb{E} \left[ \log(D_\theta(X^{(0)}, X^{(1)}, \ldots, X^{(k)})) \right]. \quad (23)
$$

where we again used the fact that $1 - D_\theta(a_0, a_1, \ldots, a_k) = D_\theta(a_k, a_{k-1}, \ldots, a_0)$. Now, substituting the definition of $D_\theta$ form (17), the gradient of (23) can be expressed as

$$
\begin{aligned}
\nabla_\theta \ell_{\text{chain}} &= \mathbb{E} \left[ \nabla_\theta \log \left( 1 + \prod_{i=1}^k w_\theta(X^{(i-1)}, X^{(i)}) \right) \right] \\
&= \mathbb{E} \left[ \left( 1 + \prod_{i=1}^k w_\theta(X^{(i-1)}, X^{(i)}) \right)^{-1} \nabla_\theta \left( \prod_{i=1}^k w_\theta(X^{(i-1)}, X^{(i)}) \right) \right] \\
&= \mathbb{E} \left[ \left( \frac{1 + \prod_{i=1}^k w_\theta(X^{(i-1)}, X^{(i)})}{\prod_{i=1}^k w_\theta(X^{(i-1)}, X^{(i)})} \right)^{-1} \frac{\nabla_\theta \left( \prod_{i=1}^k w_\theta(X^{(i-1)}, X^{(i)}) \right)}{\prod_{i=1}^k w_\theta(X^{(i-1)}, X^{(i)})} \right] \\
&= \mathbb{E} \left[ \left( 1 + \prod_{i=1}^k w_\theta(X^{(i-1)}, X^{(i)})^{-1} \right)^{-1} \nabla_\theta \log \left( \prod_{i=1}^k w_\theta(X^{(i-1)}, X^{(i)}) \right) \right] \\
&= \mathbb{E} \left[ \alpha_\theta(X^{(0)}, \ldots, X^{(k)}) \left( \nabla_\theta \log p_\theta(X^{(k)}) - \nabla_\theta \log p_\theta(X^{(0)}) \right) \right], \quad (24)
\end{aligned}
$$

where we used the definition of $\alpha_\theta$ from (19).

## D    TOY EXPERIMENT AND TRAINING DETAILS

We here describe the full details of the toy model and learning configuration, which we used to produce the results in the paper. The code for reproducing the results is available at —- (for the blind review the code will be available in the supplementary material).

The toy model used in the paper consists of a distribution concentrated around a 2D spiral embedded in a 10 dimensional space. Denoting the 10 orthogonal axes of the standard basis in this space by

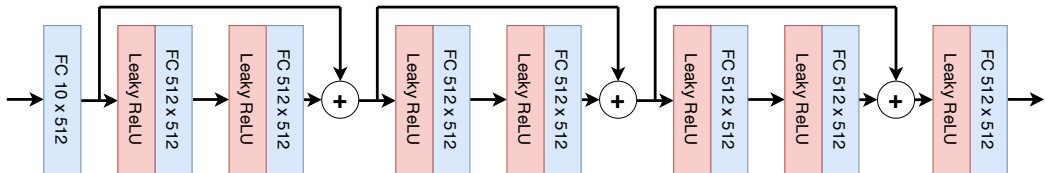

Figure 5: The architecture.

$e^{(1)}, \ldots, e^{(10)}$, the spiral lies in the $e^{(1)}$-$e^{(2)}$ plane and is confined to $[-1, 1]$ in each of these two axes. The samples of the model are produced by selecting random points along the spiral and adding Gaussian noise to them. In order to keep the samples close to the $e^{(1)}$-$e^{(2)}$ plane we used a non-isotropic noise with an STD of $0.05$ in the $e^{(1)}$ and $e^{(2)}$ directions, and an STD of $0.01$ in the directions $e^{(3)}, \ldots, e^{(10)}$.

As a parametric model for $\log p_\theta(x)$, we used an 8-layer multi-layer perceptron (MLP) of width 512 with skip connections, as illustrated in Fig. 5.

Throughout the paper we referred to the results of five different learning configurations.

1. **CNCE with an optimal (small) variance**. This configuration uses additive Gaussian noise as its contrastive distribution. We found $0.0075$ to be the STD of the Gaussian which produces the best results.

2. **CNCE with a large variance**. This configuration is similar to the previous one except for the STD of the Gaussian which was set to $0.3$ in order to illustrate the problems of using a conditional distribution with a large variance.

3. **CD without any MCMC correction**. For the MCMC process we used 5 steps of Langevin dynamics, where we did not employ any correction for the inaccuracy which results from using Langavin dynamics with a finite step size. We found $0.0075$ to be the step size (multiplying the standard Gaussian noise term) which produces the best results.

4. **CD with MH correction**. This configuration is similar to the previous one except for a MH rejection scheme which was used during the MCMC sampling. In this case we found the step size of $0.0125$ to produce the best results.

5. **Adjusted CD**. This configuration is similar to the previous one except that we used the method from Sec. 3.4 instead of MH rejection. Similarly to the previous configuration, we found the step size of $0.0125$ to produce the best results.

The optimization of all configurations was preformed using SGD with a momentum of $0.9$ and an exponential decaying learning rate. Except for the training of the third configuration, the learning rate ran down from $10^{-2}$ to $10^{-4}$ over $100000$ optimization steps. For the third configuration we had to reduce the learning rate by a factor of 10 in order to prevent the optimization from diverging.

In order to select the best step size / variance for each of the configurations we ran a parameter sweep around the relevant value range. The results of this sweep are shown in Fig. 6.

For the selection of the number of training steps, we have iteratively increased the number of steps until the results stopped improving for all configurations. These results are presented in Fig. 7.

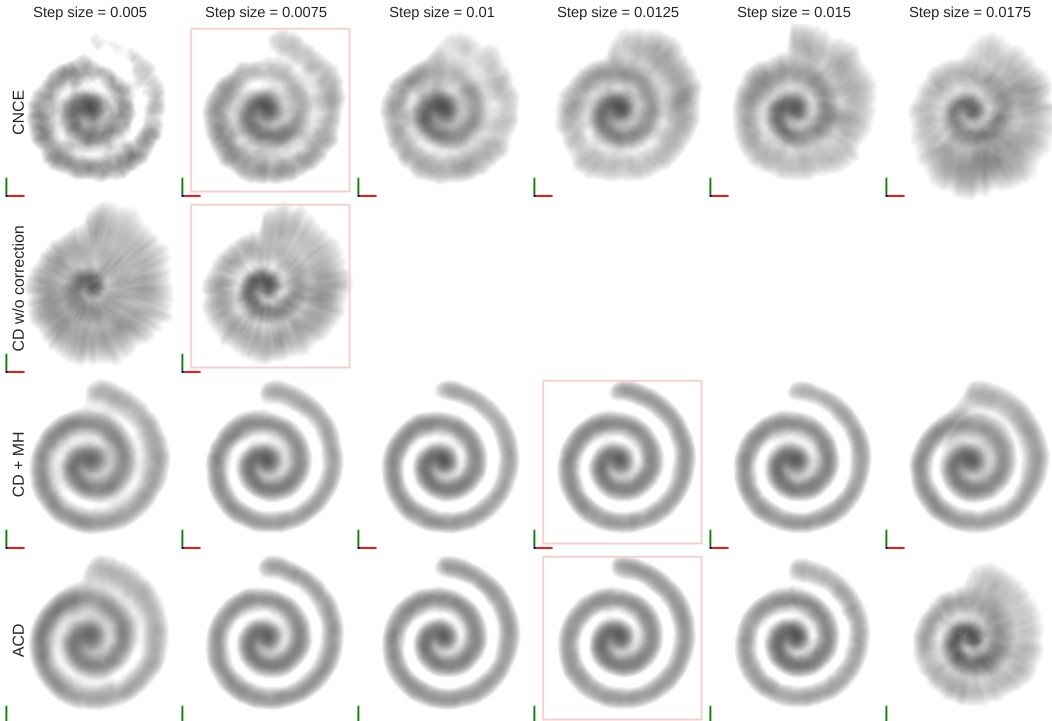

Figure 6: The step size sweep. In the case of CNCE, the step size is in fact the STD of the conditional distribution. For the case of CD, the training has diverged for large step sizes even after the learning rate has been significantly reduced. The highlighted figures indicate the selected step sizes.

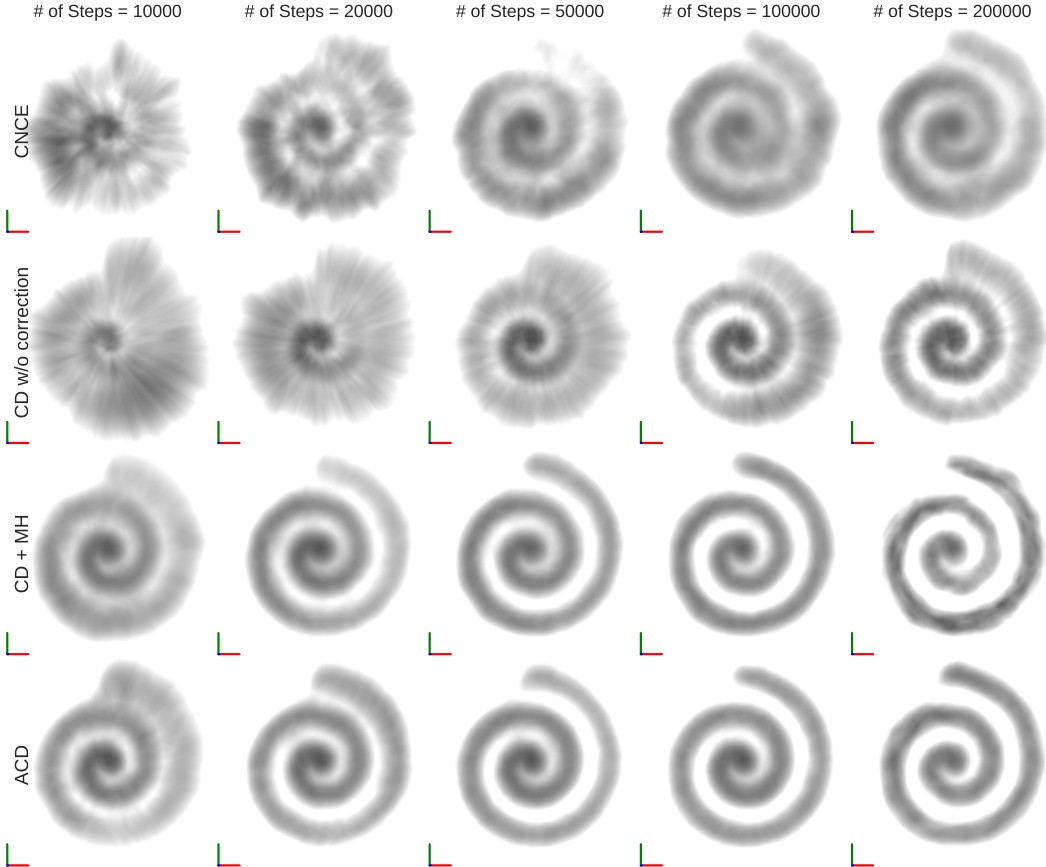

Figure 7: Selecting the number of training steps.

