# OpenReview forum: "Contrastive Divergence Learning is a Time Reversal Adversarial Game"
_ICLR.cc/2021/Conference — ICLR 2021 Spotlight_

### Official Review · AnonReviewer1 · 2020-10-25
**Interesting formulation, practical impact seems limited**

**Rating:** 6
**Confidence:** 4

**Review:**

Post-rebuttal update: Thank you for your response.  My concerns are relatively minor and I believe this work is above the acceptance threshold.

### Summary

This paper formulates CD-k training as an adversarial game, where the EBM parameterizes a discriminator which tries to classify whether a k-step Markov chain is reversed or not.

### Reasons for Score

Pros:

+ While there have been analyses of CD training under different restrictive scenarios, to my knowledge this is the first paper whose formulation applies to the CD-k algorithm used in practice.

+ The derivations seem correct.

Cons:

- While it is nice to have a justification for CD-k, the manuscript could be improved if there is more discussion about practical implications of this formulation. For example, is it possible to include a discussion on the commonly used tempered Langevin dynamics (Nijkamp et al, 2019, Du & Mordatch, 2019), which seems justifiable with the current formulation?

- The impact of the work is still limited by the fact that it doesn't explains the PCD algorithm, which is more common in large-scale settings.

### Minor Comments

- It is probably better to change the notations to signify the fact that the gradient of $\tilde{X}$ or $X^{(k)}$ wrt $\theta$ will not be accounted, e.g., by using $\partial_\theta$ instead of $\nabla_\theta$, or $\text{stop\_gradient}$.

- It is technically incorrect to say that CD's update steps don't correspond to the gradient of any objective. CD-1 with infinitesimal step-size corresponds to various well-defined objectives, see the references below.

- In Eq.(15), the subscript in $E_{q_\theta}$ should probably be dropped, to be consistent with the rest of the paper.

### References

Hyvarinen, Aapo. "Connections between score matching, contrastive divergence, and pseudolikelihood for continuous-valued variables." IEEE Transactions on neural networks 18.5 (2007): 1529-1531.

Sohl-Dickstein, Jascha, Peter Battaglino, and Michael R. DeWeese. "Minimum probability flow learning." arXiv preprint arXiv:0906.4779 (2009).

---

> ### Author Response · Authors · 2020-11-20
> **Thanks for your time and feedback.**
>
> The goal of this paper was to answer a long ongoing debate regarding the validity of the CD algorithm by suggesting an alternative way to view the algorithm’s objective. The immediate impact is indeed limited but we hope that this alternative derivation will improve the intuition regarding the learning process of CD and possibly inspire new variants of the algorithm.
>
> - **CD’s updates are gradients of some objective**: You are indeed correct that in the limit where the step size approaches zero, the update step of CD converges to a gradient of some fixed loss function. This is shown in the paper by Hyvarinen, which you mentioned, and we have therefore corrected the first sentence in the last paragraph of section 2.1 to make the statement more accurate. Note, however, that this is generally incorrect for non-infinitesimal step sizes, as shown in the references we cite. Regarding the paper by Sohl-Dickstein et al., which you mentioned, they don’t show the equivalence of their method to CD. They only show the updates of the two methods are similar (see their Sec. 3.1) but not exactly the same. In any case, we added the first reference to our corrected statement.
> - **The PCD algorithm**: It would certainly be very interesting to analyze the commonly used PCD algorithm. However, this would probably require a quite different approach than what we used here since, as opposed to what its name suggests, this algorithm is not really a variant of CD. PCD is actually an efficient version of the approximated maximum likelihood method, in which the MCMC chain is assumed to converge to the model’s distribution (and it does not impose any restrictions on the chains’ initialization). Since our derivation strongly relies on the Markov chains to be initialized using samples from the true distribution, we did not find a simple way to extend our derivation to include the approximated maximum likelihood algorithm.
> - **Justifying the use of tempered Langin Dynamics in [1,2]**: Again, we share your view that this is an important research direction. But unfortunately, following this last point, since [1] uses PCD and [2] uses a version of approximated maximum likelihood with a limited number of MCMC steps (not initialized from the dataset), we currently can’t use our analysis to justify their use of tempered Langevin dynamics.
> - **Notation highlighting that the derivative according to $\tilde{X}$ is not accounted for**: This is indeed an important point in our derivation. Following your comment (and remarks by the other reviewers), we added an extra paragraph at the end of Section 3.4 in order to try and make this point clearer. We tend to believe that use of the partial derivative notation will only make the equation more cluttered and will not necessarily make this issue clearer.
> - **Subscript below the expectation value in Eq. (15)**: We assume you’re referring to the subscript $\theta$ in $q_\theta$. We removed it, thanks!
>
>   In case you were referring to the entire subscript of the expectation (indicating the distributions of the random variables), our convention was to have it in all expectations except for those with the long chains, which involve a large number of random variables (including it in those expectations would make the equations cluttered).
>
> #### References
>
> [1] Yilun Du and Igor Mordatch.  Implicit generation and generalization in energy-based models.  InAdvances in Neural Information Processing Systems, volume 32, pp. 3608–3618, 2019.
>
> [2] Erik  Nijkamp,  Mitch  Hill,  Song-Chun  Zhu,  and  Ying  Nian  Wu.   Learning  non-convergent  non-persistent short-run mcmc toward energy-based model.  InAdvances in Neural Information Pro-cessing Systems, pp. 5232–5242, 2019.

---

### Official Review · AnonReviewer4 · 2020-10-28
**This paper is acceptable, and the ICLR community may benefit from the contributions this paper brings to light.**

**Rating:** 7
**Confidence:** 3

**Review:**

Summary:
This paper presents a way to view contrastive divergence (CD) learning as an adversarial learning procedure where a discriminator is tasked with classifying whether or not a Markov chain, generated from the model, has been time-reversed. Beginning with the classic derivation of CD and its approximate gradient, noting relevant issues regarding this approximation, the authors present a way to view CD as an extension of the conditional noise contrastive estimation (CNCE) method where the contrastive distribution is continually updated to keep the discrimination task difficult. Specifically, when the contrastive distribution is chosen such that the detailed balance property is satisfied, then the CNCE loss becomes exactly proportional the CD-1 update with the derivation further extended to CD-k. Practical concerns regarding lack of detailed balance are mitigated through the use of Metropolis-Hastings rejection or an adaptive weighting that arises when deriving the gradient of their time-reversal classification loss.  A toy example providing empirical support for correcting the lack of detailed balance is included.

Strengths:
The paper is well written. From "first principles" through the CD-CNCE link, the paper was straightforward to follow without technical issues and with appropriate references. The results of this work appear novel, and proofs seem correct. The ability to use the weighting to address detailed balance in practice is neat, and the experiments, though limited, show promise.

Concerns:
I understand performance comparisons and experiments were not the focus of the paper. However, considering the newly presented link between CNCE and CD, it would have been exciting to see some evaluation metrics. Perhaps even just a simple experiment from the original NCE or CNCE papers.
For the MCMC process, appendix D mentions that 5 steps of Langevin dynamics were used. How was 5 selected? Was there any significant gain or degradation when it varied? More generally, what kind of impact does chain length have on the discriminator’s classification ability? Does chain length affect the behavior of CD with MH correction and adjusted CD similarly?

Minor:
In Figure (3b), it is said that CD is based on Langevin dynamics MCMC adjusted with the method of Sec. 3.4 yet both MH and the weight adjustment are included there. Which one is in Fig. (3.4)

---

> ### Author Response · Authors · 2020-11-20
> **Thank you for your time and your feedback.**
>
> - **Evaluation**: Kindly note that we have intentionally tried to avoid performing a thorough comparison between CNCE and CD since this is not the goal of this paper. Our main goal was to show that CNCE is a framework, which can be generalized to include CD as a special case. The purpose of the toy model was only to shed light on the importance of allowing the conditional distribution $q$ to adapt to the learned model. The model was therefore explicitly designed to be challenging for the classic CNCE. As for redoing the experiments in the original NCE and CNCE papers, note that they only performed a simple ICA experiment and illustrated results on some very simple 2D toy examples (simpler than our 10D example). We therefore don’t believe that repeating these experiments would benefit this work.
> - **Selection of k**: This is a good point. k was intentionally selected to be small in order to mimic the behavior of the algorithms for complex (possibly high-dimensional) distributions. In such cases, k is usually extremely small relative to the number of steps it takes for the MCMC process to converge to the model's distribution. When increasing k, the performance of all CD based methods improve. It’s important to note however that for a larger k the optimal step size will be different, usually smaller.
> - **Difference between MH and ACD as a function of k**: One noted difference between the adjusted version of CD and the MH corrected version is that for large values of k, the adjusted version has usually resulted in a slightly smaller step size relative to the MH version (the step size is automatically adjusted to keep the average weight value constant). But since this method doesn’t reject any steps from the Markov chain, the overall distances traversed by the chains were in fact larger. In any case, we weren’t able to observe any differences in performance between the two methods on this model or on others.
> - **Fig 3b**: Thanks for pointing this out. We have used the adjusted version of CD in this figure. We have updated the caption to make it clearer. The selection of the adjusted version over the MH version was arbitrary as both methods have produced very similar results.

---

### Official Review · AnonReviewer2 · 2020-10-29
**Establishes an interesting theoretical link**

**Rating:** 7
**Confidence:** 3

**Review:**

Summary

To implement the contrastive divergence (CD) algorithm in practice, an intractable term is typically omitted from the gradient. This leads to an approximation. This work shows that the resulting algorithm can in fact be viewed as an exact algorithm targeting a different, adversarial objective. The derivation in this paper also shows how Markov chains which are not reversible w.r.t. the posterior distribution of interest can be employed within the algorithm. Effectively, this assigns an importance weight to each sample which akin to the acceptance ratio which would be needed for a Metropolis--Hastings type correction.



Strengths and novelty

To my knowledge, the derivation of the relationship between CD and conditional noise contrastive estimation (CNCE) is novel. Making it clear how these algorithms are related is a contribution worth publishing.


Weaknesses

Perhaps having an additional non-toy example would have been a nice illustration. However, since the paper's main focus is on establishing theoretical connections between CD and CNCE, I do not believe that the lack of further numerical examples should preclude publication.



Minor comments

- A number of entries in the bibliography have typos such as missing capitalisation of proper nouns inconsistent use of capital letters in journal and conference names.

- I don't understand the last term in Eq. 2. This needs to be more rigorously written.

- Section 3.2 extablishes that the CD-1 gradient is a special case of the CNCE gradient. This would mean that both CD-1 and CNCE lead to computationally the same algorithm. However, this appears not to be the case in the toy example. For readers not familiar with CNCE, please make it more clear in what way CD-1 and CNCE differ in practice if both use the same reversible Markov chain.

- In Section 2.1, it is stated that CD does not use log-likelihood loss. However, it seems to me that for $p_\theta$-reversible Markov chains, if the chain is either fast mixing or the number of iterations $k$ sufficiently large, the gradient in Eq. 3 is proportional to the gradient of the log-likelihood (multiplied by $-1$) in expectation because in this case, the first term has expectation zero.

---

> ### Author Response · Authors · 2020-11-20
> **Thanks for the comments and good suggestions.**
>
> - **Last term in eq. (2)**: We have chosen to write this term in the same way it appeared in the original paper [1]. This term refers to the gradient of the loss only w.r.t. the $\theta$ that appears within $r_\theta^k$. Following your comment, we added an explanation with an alternative way to write this term.
> - **Difference between the CNCE and CD-1**: The main difference between the two is that CNCE uses a fixed conditional distribution $q$ (that doesn’t change from one step to the next), while CD-1 uses a MCMC transition rule that depends on the learned model and therefore changes over time. Please see Section 3.2 (From CNCE to CD-1), which describes the transition from the fixed $q$ to the adaptive one, and the illustration of the difference in Figure 2. Following your question, we added an extra sentence at the beginning of section 3.2 to make it clearer that CNCE uses a fixed $q$.
> - **Connection to gradients of the log-likelihood**: It is correct that as k becomes extremely large the CD-k loss converges to the maximum-likelihood loss, but this convergence usually requires extremely large values of k. For practical values of k (tens or hundreds) the two will usually significantly differ from one another. The best way to see this convergence is by looking at the CD-k loss in Eq. (1). Here, as k goes to infinity $r^k_{\theta}$ convergence to $p_{\theta}$ and the second term cancels out. This leaves us with the Kullback Leibler divergence as the loss function (which is equivalent to maximizing the likelihood).
> - **Bibliography**: We fixed the errors in the bibliography. Thanks.
>
> #### References
>
> [1] Hinton, Geoffrey E. "Training products of experts by minimizing contrastive divergence." Neural computation 14.8 (2002): 1771-1800.

---

### Official Review · AnonReviewer3 · 2020-10-29
**A new theoretical understanding of contrastive divergence as adversarial training**

**Rating:** 8
**Confidence:** 4

**Review:**

# General statements
This paper has a special flavour, in the sense that it provides new light on a very established training method for energy-based models: contrastive divergence. Its core contribution is to provide a theoretically grounded understanding of CD as it is widely used, avoiding the common assumption that this algorithm stems out of a simplifying assumption.

This is done through a connection between CD and adversarial training. On their way, the authors show how some minor corrections suggested by their interepretation may dramatically improve performance of CD, at least on their toy example.

Since CD is a widely accepted method, I feel that the deliberate choice of restricting their experiments on toy data is legitimate.


All in all, I would say that the paper is a very nice read, and its english usage is good, as well as the references that are appropriate.
I think that it is appropriate for presentation at ICLR, since it may stimulate new research on CD.


# Detailed comments
Below are some minor comments in chronological order
## Introduction
* "Thus, Our": uppercase

## Toy example
* In figure 4, you probably mean "from left to right"
* To be extra sure, are you effectively disabling gradient recording when computing \tilde{x} as I assume you do ? I'm asking because \tilde{x} actually appears as a function of x, parameterized by \theta, i.e. as \tilde{x}_\theta(x), since it involves the transition kernel q_theta for its computation. As you write below eq. (17), you are considering that the kernel q as kept fixed, explaining such a choice.
however, and if I'm not mistaken, it should not be too difficult with autograd mechanics to include this dependency in the updates. Did you try it ? Did it break the algorithm ?
* I would appreciate more steps in your derivations (22) and (24): I don't follow easily the transitions to lines 2 and 4 of each.
* The neural net used for the toy data looks impressively large (8 layers of FC+leakyReLU with 512 hidden size). Was it really necessary ?

---

> ### Author Response · Authors · 2020-11-20
> **Thanks for the good points you raised and the positive feedback.**
>
> - **Gradient w.r.t. $\tilde{x}$**: This is an important point. You are indeed correct regarding the fact that the gradients according to $\tilde{x}$ are not taken into account when updating the probability model. This choice is not made for convenience. It would simply be incorrect to take into account q when updating the probability model. When generating the Markov chains, $q_{\theta}$ is being used in order to produce chains that appear to be time-reversible. During the update stage, we try to adjust the model so that it will be better in classifying whether the chains were reversed. Therefore we do not want the optimization to affect the generation process. This is just like in GANs, where the generator and discriminator have different objectives, and so when updating the discriminator, one doesn’t take into account how the generator would change in response. In our case, the transition rule $q_{\theta}$ acts as a generator which generates the contrastive samples, and the probability model $p_\theta$ is part of the discriminator. We added this discussion to the end of section 3.2 in order to try and make it clearer.
> - **The architecture in the toy example**: Since the experiment itself was not the main focus of the paper, we did not devote a lot of effort to optimizing the architecture and it is very likely that a much smaller one would have been sufficient. We have initially started with the network that was used in [1] as a discriminator of GAN that was used to learn a 2D spiral dataset. This network uses 5 FC+leakyReLU layers of width 512. In our setting, we have encountered stability issues training this network which we believe to have resulted from the fact that we didn’t use any normalization layers. We have found that by replacing each FC layer with a Residual block of 2 FC layers we can make the network significantly more stable. That led us to the resulting architecture. It is also important to keep in mind that our toy model is slightly more challenging than the simple 2D spiral due to its embedding in a 10-dimensional space.
> - **Typos and additional details**: Thanks. We corrected the errors you pointed out and added a few more intermediate steps to the derivation in eqs. (22) and (24).
>
> #### References
>
> [1] Tanaka, Akinori. "Discriminator optimal transport." Advances in Neural Information Processing Systems. 2019.

---

### Public Comment · ~Jianwen_Xie1 · 2020-11-14
**related works about deep EBMs**

Dear Authors and Reviewers,

We found that the current paper missed some important references about pioneering works that are related to energy-based generative models parameterized with deep net energy.

The first paper that proposes to train an energy-based model parameterized by modern deep neural network and learned it by Langevin based MLE is in (Xie. ICML 2016) [1]. The model is called generative ConvNet, because it can be derived from the discriminative ConvNet. This is also the first paper to formulate modern ConvNet-parametrized EBM as exponential tilting of a reference distribution, and connect it to discriminative ConvNet classifier. That is, EBM is a generative version of a discriminator. (Xie. ICML 2016) [1] originally studied such an EBM model on image generation theoretically and practically in 2016.

(Xie. CVPR 2017) [2] (Xie. PAMI 2019) [3] proposed to use Spatial-Temporal ConvNet as the energy function in EBMs for video generation. The model is called Spatial-Temporal generative ConvNet.

[2] and [3] are the first paper to give adversarial interpretation of maximum likelihood learning of ConvNet-EBM, That is the EBM serves the roles of both the generator (actor) and the discriminator (critic). The MLE learning is self-critic.

(Xie. CVPR 2018) [4] also proposed to use volumetric 3D ConvNet as the energy function for 3D shape pattern generation. It is called 3D descriptor Net.

Also, the Generative Cooperative Nets (CoopNets) (Xie. PAMI 2018)[5] and (Xie. AAAI 2018) [6], which jointly trains an EBM and a generator network by MCMC teaching.

Those are the more original and earlier papers for deep EBMs with ConvNet as energy function than what you have cited, e.g., [7](Yilun Du and Igor Mordatch, 2019).

References:

[1] A Theory of Generative ConvNet. Jianwen Xie *, Yang Lu *, Song-Chun Zhu, Ying Nian Wu (ICML 2016)

[2] Synthesizing Dynamic Pattern by Spatial-Temporal Generative ConvNet Jianwen Xie, Song-Chun Zhu, Ying Nian Wu (CVPR 2017)

[3] Learning Energy-based Spatial-Temporal Generative ConvNet for Dynamic Patterns Jianwen Xie, Song-Chun Zhu, Ying Nian Wu IEEE Transactions on Pattern Analysis and Machine Intelligence (TPAMI) 2019

[4] Learning Descriptor Networks for 3D Shape Synthesis and Analysis Jianwen Xie *, Zilong Zheng *, Ruiqi Gao, Wenguan Wang, Song-Chun Zhu, Ying Nian Wu (CVPR) 2018

[5] Cooperative Training of Descriptor and Generator Networks. Jianwen Xie, Yang Lu, Ruiqi Gao, Song-Chun Zhu, Ying Nian Wu. IEEE Transactions on Pattern Analysis and Machine Intelligence (TPAMI) 2018

[6] Cooperative Learning of Energy-Based Model and Latent Variable Model via MCMC Teaching. Jianwen Xie, Yang Lu, Ruiqi Gao, Ying Nian Wu. AAAI 2018.

[7] Yilun Du and Igor Mordatch. Implicit generation and modeling with energy based models. In Advances in Neural Information Processing Systems, pages 3603–3613, 2019

Thank you!

---

> ### Author Response · Authors · 2020-11-21
> **Thank you for referring us to these papers as they present good examples for using energy based models.**
>
> Since our discussion is agnostic to the actual implementation of the energy-based model, we did not dive into this topic and have only provided a handful of examples in order to establish the motivation for using such models. Specifically, in the context of neural networks, we have chosen to bring some of the most recent works in the field rather than provide an exhaustive survey. Since [1] presented a breakthrough in using energy-based models with deep nets, we have updated our paper and added this reference at the end of the first paragraph as an additional example.
>
> #### References
>
> [1] A Theory of Generative ConvNet. Jianwen Xie *, Yang Lu *, Song-Chun Zhu, Ying Nian Wu (ICML 2016)

---

### Decision · Program_Chairs · 2021-01-07
**Final Decision**

**Decision:**

Accept (Spotlight)

**Comment:**

This paper reveals a novel interpretation of the well-established CD for energy-based model training as an adversarial game through conditional NCE. The paper could be potential impactful for the community of EBMs.

There are several points should be addressed in final version:

1, Based on such an interpretation, the number of steps becomes a tunable parameters, rather than in vanilla understaning in CD-family (the larger, the better in terms of approximation, by with more computation cost).

2, It is okay to stop the gradient when solving an adversarial game as the paper discussed. However, propagating the gradient through the component is also another choice, which leads to the algorithm proposed in [1].

It will be interesting to discuss these in the paper.

[1] Sohl-Dickstein, Jascha, Peter Battaglino, and Michael R. DeWeese. "Minimum probability flow learning." arXiv preprint arXiv:0906.4779 (2009).